# A Novel Linear B-Cell Epitope on the P54 Protein of African Swine Fever Virus Identified Using Monoclonal Antibodies

**DOI:** 10.3390/v15040867

**Published:** 2023-03-28

**Authors:** Nannan Zheng, Chao Li, Haoyu Hou, Yinlong Chen, Angke Zhang, Shichong Han, Bo Wan, Yanan Wu, Hua He, Na Wang, Yongkun Du

**Affiliations:** 1College of Veterinary Medicine, Henan Agricultural University, Zhengzhou 450046, China; 15517237872@163.com (N.Z.); lc20230301@163.com (C.L.); hhy8417@gmail.com (H.H.); cyl200016@163.com (Y.C.); zhangangke1112@126.com (A.Z.); hanshichong081@126.com (S.H.); wanboyi2000@163.com (B.W.); wlyananjiayou@yeah.net (Y.W.); hehua1123@126.com (H.H.); na-wang@163.com (N.W.); 2International Joint Research Center of National Animal Immunology, College of Veterinary Medicine, Henan Agricultural University, Zhengzhou 450046, China; 3Henan Engineering Laboratory of Animal Biological Products, Zhengzhou 450046, China

**Keywords:** African swine fever virus, P54, vaccine development, B-cell epitope, monoclonal antibody

## Abstract

The African swine fever virus (ASFV) is a highly infectious viral pathogen that presents a major threat to the global pig industry. No effective vaccine is available for the virus. The p54 protein, a major structural component of ASFV, is involved in virus adsorption and entry to target cells and also plays a key role in ASFV vaccine development and disease prevention. Here, we generated species-specific monoclonal antibodies (mAbs), namely 7G10A7F7, 6E8G8E1, 6C3A6D12, and 8D10C12C8 (subtype IgG1/kappa type), against the ASFV p54 protein and characterized the specificity of these mAbs. Peptide scanning techniques were used to determine the epitopes that are recognized by the mAbs, which defined a new B-cell epitope, TMSAIENLR. Amino acid sequence comparison showed that this epitope is conserved among all reference ASFV strains from different regions of China, including the widely prevalent, highly pathogenic strain Georgia 2007/1 (NC_044959.2). This study reveals important signposts for the design and development of ASFV vaccines and also provides critical information for the functional studies of the p54 protein via deletion analysis.

## 1. Introduction

The African swine fever (ASF) caused by the African swine fever virus (ASFV) is a highly infectious disease with a high mortality rate. Pigs of all breeds and different ages are susceptible to ASF. The disease is characterized by a febrile syndrome with erythema and cyanosis of the skin; the attenuation of internal organ functions, especially those of the digestive system, with symptoms such as vomiting and hemorrhagic diarrhea; and depression. Anorexia, cyanosis, and incoordination occur 1–2 days before death [1,2,3]. Visceral lesions are mainly manifested by splenomegaly, hemorrhage in organs and visceral lymph nodes, and excess natural fluid in body cavities and spaces. ASF was first detected in Kenya in 1909 and was subsequently found in Central and West Africa. The disease was limited to sub-Saharan African countries until 1957 when it was identified in Portugal and other regions of Africa [4,5,6]. The first case of ASF was discovered in China in the northern district of Shenyang in 2018 [7]. No vaccine or specific treatments are available for the condition. Therefore, disease control is mainly based on early detection and the application of strict health and biosecurity measures [8,9]. The epidemiology of ASF is complex because of diverse viral transmission routes. Moreover, surviving pigs may remain infected for months, which may lead to extended virus transmission and thus to the spread and maintenance of the disease, which further complicates efforts to control pathogenesis [10]. Therefore, there is an urgent need for localized diagnostic reagents with fully independent intellectual property rights that help prevent and control the ASF epidemic in China and elsewhere.

ASFV, the only member of the genus, is a complex enveloped virus with an icosahedral morphology [11] that comprises four concentric layers and a double-stranded DNA molecule. The viral genome is 170–193 kb [11,12] and consists of a conserved central region of 125 kb with non-complementary variable regions at both ends. Differences in genome length among ASFV isolates are mainly due to the gain or loss of members of multigene families in the left and right variable regions [13,14]. The ASFV genome contains 151 to 167 open reading frames that encode 54 structural proteins and approximately 100 polypeptides that mediate targeting to monocytes and macrophages [15]. Among these proteins, p30, p54, p72, and the polyprotein pp62 are antigenic proteins that mediate the induction of antibodies [16,17]. However, despite the utility of these factors as serodiagnostic targets, these proteins have not yet been used for the development of antibody-mediated protection against viral strains [18].

Monoclonal antibodies (mAbs) against the major ASFV shell protein p72 and the structural hyperimmunogenic protein p30 have been used as targets in serological assays to detect ASFV infection [19,20]. The p54 protein, a type II transmembrane protein that is produced following infection, is also a potentially important target for the detection of ASFV [21,22]. Moreover, p54 inoculation induces high titers of anti-p54 antibodies [23,24]. Furthermore, anti-p54 antibodies appear as early as 10 days after infection and persist in the blood for several weeks [25,26]. Therefore, serological tests against p54 may be effective for ASFV detection. Here, we define a new B-cell epitope of p54 that is conserved among all reference ASFV strains from different regions of China. This novel epitope has the potential for the design and development of new ASFV vaccines that combat infection by this important and widespread swine pathogen.

## 2. Materials and Methods

### 2.1. Genes, Cell Lines, and Serum

The gene sequence for the p54 protein was obtained from NCBI (GenBank accession number MK128995.1, 1-399bp). Based on the p54 gene sequence, the pMD-18T-E183L plasmid was synthesized by DongXuan Gene Technology Co., Ltd. (Kunshan, China). Marc-145 and human embryonic kidney 293T (HEK293T) cell lines were obtained from BasalMedia (Shanghai, China) and were cultured in DMEM (Gibco-BRL) with 10% (*v*/*v*) heat-inactivated FBS (Gibco-BRL). Myeloma SP2/0 cells obtained from BasalMedia were cultured using HybGro™ Hybridoma cell serum-free medium III without the addition of fetal bovine serum. Cells were cultured at 37 °C in a humidified incubator (Thermo-Fisher Scientific, Waltham, MA, USA) containing 5% CO_2_. In order to facilitate protein expression, the 53 *N*-terminal hydrophobic amino acids of the p54 protein were truncated, and the remaining fragment was retained. The truncated p54 gene sequence (GenBank accession number MK128995.1) was directly cloned into the pET-30a(+) vector after digestion with restriction enzymes NdeI and HindIII and ligation with T4 DNA ligase purchased from New England Biolabs. *Escherichia coli* DH5α competent cells were purchased from TransGen Biotech (Beijing, China). The anti-ASFV sera were obtained from the China Veterinary Culture Collection Center (Beijing, China).

### 2.2. Expression and Purification of Recombinant p54 Protein

The p54 gene was amplified with the upstream primer E183L-F and the downstream primer E183L-Rev (Songon Biotech, Shanghai, China) using the synthetic pMD-18T-E183L plasmid as a template. The amplified product and pET-30a(+) vector were cleaved with NdeI and HindIII followed by T4 DNA ligase treatment. The recombinant plasmid was identified as pET-30a(+)-p54 and the integrity of the cloned gene was verified via sequencing. Plasmid pET-30a(+)-p54 was transformed into *E. coli* BL21 (DE3) and cultured on LB agar plates overnight at 37 °C. The transformants were picked into an LB medium containing kanamycin for plasmid selection. IPTG inducer was added to a final concentration of 0.1 mmol/L followed by protein induction for 6 h. Cells were collected, resuspended in PBS, lysed by sonication, and centrifuged. The supernatant and pellet fractions were collected and analyzed via SDS-PAGE to detect p54 protein expression. The protein was purified from the supernatant fraction using Ni-affinity chromatography. Preloaded column and elution fractions were collected for SDS-PAGE and Western blot analyses.

### 2.3. Animal Immunization Strategies and Preparation of Monoclonal Antibodies

Female BALB/c mice, aged 6 to 8 weeks old, were obtained from the Animal Experimental Center of Huaxing in Zhengzhou, China. The mice were kept in specific pathogen-free (SPF) isolators with negative pressure and had ad libitum access to food and water. The purified p54 recombinant protein was mixed with an equal volume of Freund’s adjuvant (Sigma, Merck KGaA, Darmstadt, Germany) and emulsified, and a subcutaneous immunization dose of 100 μg/200 μL was introduced per BALB/c mouse. The first immunization with Forster’s complete adjuvant was followed by second and third immunizations with Forster’s incomplete adjuvant at two-week intervals. Blood was collected one week after the third immunization, and serum antibody potency was assessed using indirect ELISA. Mice with the highest potency were selected for superimmunization. To select hybridoma cells secreting mAbs against the p54 protein, cell fusion was performed three days after superimmunization. Single-cell suspensions were prepared from mouse spleens after three days of superimmunization, and cell fusion was performed with SP2/0 cells at a 10:1 ratio by PEG 1500 action. The fused cells were seeded into 96-well plates containing a culture medium supplemented with hypoxanthine–aminopterin–thymidine (HAT) to select for hybridomas. The plates were incubated at 37 °C in 5% CO_2_. On days 7 and 9, the HAT-containing medium was replaced with a fresh medium. The fused cells were screened via indirect ELISA, positive hybridoma cells were picked, and the cells were subcultured three times using the limited dilution method. The p54-protein-specific monoclonal cell line was chosen, expanded, and then frozen.

### 2.4. Monoclonal Antibody Characterization

Mice superimmunized with purified p54 recombinant protein were prepared as described in the preceding section. Mice were euthanized after three days of superimmunization, and the spleens were taken, ground, and prepared into single-cell suspensions. SP2/0 single-cell suspensions with good growth were selected and mixed with the spleen cells (SP2/0:splenocytes = 1:10), centrifuged, and mixed with a GNK buffer (0.8% NaCl, 0.04% KCl, 0.2% glucose, 0.001% phenol red, 0.356% Na_2_HPO_4_·12H_2_O, and 0.078% NaH_2_PO_4_·2H_2_O). Then, 1 mL of pre-warmed PEG 1500 was added dropwise at a rate of 1 mL/min to the mixed cells to promote cell fusion. A HAT medium was used to culture the fused cells in 96-well plates at 37 °C under 5% CO_2_. Half of the HAT medium was changed after four days. The fused cells were screened using indirect ELISA one week later, and positive hybridoma cells were picked. Three cell subclones were treated using the limited dilution method in order to obtain purified mAbs. The p54-protein-specific monoclonal cell line was picked, expanded, and frozen.

Western blotting was performed to determine the reactivity of mAbs with p54 protein following SDS-PAGE. The samples were wet-transferred to a PVDF membrane, which was then blocked with 5% skimmed milk powder for 2 h at room temperature. The membrane was washed three times for 10 min with PBST and then was incubated with p54 primary mAb for 1 h at 37 °C followed by three washes. HRP-conjugated goat anti-mouse IgG (Solarbio, Beijing, China) was added as a secondary antibody for 1 h at 37 °C, and membranes were washed three times, followed by enhanced chemiluminescence (NCM Biotech, Suzhou, China) color development. SP2/0 cultures were used as negative controls, and positive anti-ASFV sera (China Veterinary Culture Collection Center) were used as positive controls in these experiments.

### 2.5. Indirect Immunofluorescence Assay

The specificity of the mAbs for the p54 protein was examined using indirect immunofluorescence assays. The recombinant plasmid pcDNA3.1-NTD was transfected using Lipo-fectamine™ 2000 (Invitrogen, Carlsbad, CA, USA) into HEK293T cells in DMEM with 10% FBS in 24-well plates to a density of 70–80%. The empty vector and positive serum were used as negative and positive controls, respectively. After transfection for 24 h, the supernatant was discarded, and the cells were fixed with paraformaldehyde (4%) for 30 min or overnight at 4 °C. The cells were washed five times with PBS. Triton X-100 (0.1%) (Sigma, Shanghai, China) was added for 15 min followed by five washes with PBS. Then, 2.5% BSA (200 μL) was added per well for 1.5 h followed by five washes with PBS. Samples were incubated for 1 h at room temperature with anti-p54 mAbs as the primary antibody and then washed five times with PBS. FITC-conjugated goat anti-mouse IgG (H + L, Proteintech) was used as a secondary antibody followed by five washes with PBS. Cells were colored by antifading Mounting Medium (with DAPI) (Solarbio, Beijing, China), and viewed via fluorescence electron microscopy.

### 2.6. Design, Identification, and Screening of p54 Peptides

Nine truncated peptides were designed based on the amino acid sequence of the p54 protein. Western dot blots were performed to identify the peptides. Each peptide was spotted (1 μL) on a nitrocellulose membrane at a concentration of 1 mg/mL peptide, with the p54 protein as a positive control. The membrane was blocked for 2 h at 37 °C and then incubated with the anti-p54 mAb prepared in this study. The antibody was purified and labeled with horseradish peroxide at a dilution of 1:2000 for 1 h. The membrane was washed five times with PBST. Enhanced chemiluminescence reagents were used to detect the binding capacity of the mAb to the target peptides. This analysis identified five mAbs against three peptides: P1, P8, and P9. As the P8 peptide was not previously studied by other laboratories, this peptide was selected for further characterization. Ten additional peptides were designed (Figure 1C). Dot blot analysis revealed six peptides that reacted with p54 mAb, namely P1, P2, P3, P4, P6, and P7, which allowed for the determination of the minimal epitope recognized by this mAb.

### 2.7. Analysis of the Spatial Structure of Epitopes

PyMOL software (version 2.5.2, DeLano Scientific LLC, San Carlos, CA, USA) was used to visualize the spatial distribution of the epitopes identified by the mAbs prepared in this study.

### 2.8. Statistical Analysis

Data are expressed as means ± standard deviations. Student’s *t*-test was used for statistical comparisons. The differences were set to be significant at * *p* < 0.05 and very significant at ** *p* < 0.01 and *** *p* < 0.001.

Statistical data were analyzed using GraphPad Prism 9 software. Experimental data are presented as means ± standard deviations (SDs). Statistical analyses were performed using a *t*-test, and differences were considered significant at *p* < 0.05.

### 2.9. Blocking/Competitive ELISA Analysis of ASFV Positive Serum

Blocking/competitive ELISA assays were performed to assess the expression of p54 epitopes that were targeted by mAbs identified in positive anti-ASFV sera (China Veterinary Culture Collection Center). The purified recombinant p54 protein was mixed with phosphate-buffered saline (pH 9.6) to a final concentration of 4 μg/mL and used to coat the 96-well plates, which were incubated for 12 h at 4 °C. The plates were washed three times with TBST and shaken dry. After blocking for 1 h at room temperature with 5% SM and washing as above, the mixed ASFV-positive serum (1:2 in 1% BSA in PBS, 50 μL/well) from the culture supernatants of hybridoma cells was added with incubation for 1 h at 37 °C. SP2/0 cell culture supernatant was used as a negative control. The mixed ASFV-positive serum (diluted 1:16 in 5% SM) was added to the coated plates, along with the culture supernatants of hybridoma cells and SP2/0 cell culture supernatant as controls. The samples were incubated for 1 h at 37 °C. Negative anti-ASFV serum was treated similarly as a negative control. The plates were washed and incubated for 1 h at 37 °C with the labeled p54 mAb HRP-6E8G8E1 diluted with 5% SM at a ratio of 1:2000. The plates were washed and incubated with 3,3′,5,5′-tetramethylbenzidine (TMB, Solarbio, Beijing, China) for 10 min at room temperature. The reaction was quenched with 3 mol/L sulfuric acid (50 μL/well). Optical density was measured at 450 nm using a Tecan 10 M multimode microplate reader. Three replicates were performed for each reaction. The absorbance values were then converted to a percentage of inhibition (PI) using the following formula: PI (%) = [1 − (OD_450_ of test sample/OD_450_ of negative control)] × 100%.

## 3. Results

### 3.1. Construction of p54 Expression Plasmid and p54 Protein Purification

The pET30a(+)-p54 recombinant expression plasmid was constructed by the insertion of the p54 gene truncated in 53 N-terminal codons in the pET30a vector. Gel electrophoresis after double digestion with NdeI and HindIII showed a target band of approximately 400 bp, which was consistent with expectations. Sequencing results further confirmed that the insert comprised the p54 gene. The production of the p54 protein in *E. coli* was induced and analyzed using SDS-PAGE, which showed the induction of a major species of approximately 19 kDa, which was absent in uninduced samples. The p54 protein was mainly expressed in the supernatant fraction (Figure 1A,B). The His-tagged protein was purified via Ni-affinity chromatography. SDS-PAGE and Western blot showed that the recombinant p54 protein was present as a single species with a purity of approximately 95% (Figure 1C).

### 3.2. Preparation of p54 Monoclonal Antibodies

The potency of the mouse serum p54 protein antibody was determined using ELISA to be 1:500,000 after two weeks of triple immunization, as described in Materials and Methods (Figure 2A). Each well included two replicate wells. Single-cell suspensions were prepared from mouse spleens three days after superimmunization and were fused to myeloma SP2/0 cells. The fused cells grew for 9–12 days to form a clonal population of cells, and screening for p54 mAbs was performed using indirect ELISA. Positive samples were subcloned using the limited dilution method, and four hybridoma cell lines capable of stably secreting p54 protein-specific mAbs were obtained after three subcloning procedures (7G10A7F7, 6E8G8E1, 6C3A6D12, and 8D10C12C8). ELISA assays were performed to determine the potency of the four selected monoclonal antibodies, and the results showed that the potency of each mAb was 1:500,000 with two replicates per well (Figure 2B).

### 3.3. Identification and Characterization of p54 Monoclonal Antibodies

The isotype analysis of the four mAbs described above revealed one IgG1 isoform and three IgG2b isoforms. The light chain type of all p54 monoclonal antibodies was kappa (Figure 3A). Western blotting showed that all four mAbs reacted strongly with the denatured p54 protein, which suggests that the antibodies specifically recognized p54 (Figure 3B). Immunofluorescence assays further confirmed the specific binding of these mAbs to the ASFV p54 protein. Fluorescence was mainly observed in the perinuclear region of HEK293T cells transfected with plasmid pCMV-3 × flag-p54 (Figure 3C).

### 3.4. Epitope Mapping of Anti-P54 Monoclonal Antibody

Nine overlapping subpeptides of p54 were synthesized (Figure 4A), and dot blots were performed to determine the binding site for each mAb. All four mAbs bound strongly to peptides P1, P8, and P9 (Figure 4B). As mAb 6E8G8E1 reacted strongly with peptide P8, we further divided this peptide into ten shorter fragments (P1–P10) to define the P8 epitope more accurately (Figure 4C). Dot blots showed that when the C-terminal Q residue was removed from P6 (TASQTMSAIENLRQ), the resulting P7 peptide (TASQTMSAIENLR) was still recognized by mAb 6E8G8E1, whereas P8–P10 that lacked the residues were not bound by the antibody. The P1 peptide comprised ASQTMSAIENLRQR. Removing the three N-terminal amino acids to generate P4 (TMSAIENLRQR) did not affect binding by the 6E8G8E1 mAb, whereas P5 (MSAIENLRQR), which lacked an additional N-terminal residue, did not react with this antibody (Figure 4D). In summary, TMSAIENLR was the smallest linear epitope of p54 that was recognized by mAb 6E8G8E1.

### 3.5. Spatial Structure and Position Analysis of p54 Epitopes

The visual identification of the p54 epitope spatial structure was performed with PyMOL software. The epitope P1 (01-SSRKKKAAAIEEEDIQFINP-20) (red), P8 (106-TASQTMSAIENLRQR-120) (red), and P9 (116-NLRQRNTYTHKDLENSL-132) (red) were located on the surface of the p54 protein, as was the smallest linear epitope 110-TMSAIENLR-118 (Figure 5A). These data further demonstrate that the identified epitopes are linear and immunodominant. The sequence comparison of 13 domestic and foreign ASFV wild strains selected from the NCBI database was performed using DNAman software to explore the extent to which the epitopes of p54 mAbs were conserved. ASFV-p54-mAbs were highly conserved in domestic strains compared with foreign strains, with 100% shared sequence similarity among the domestic strains (Figure 5B).

### 3.6. Reactivity of ASFV-Positive Serum against Anti-p54 Monoclonal Antibodies

Blocking ELISA was used to assess the efficacy of mAb 6E8G8E1 to inhibit the binding of positive anti-ASFV sera to the p54 protein. The mAb blocked the binding with an inhibition rate of >50% (Figure 6). These findings suggest that the epitopes recognized by mAb 6E8G8E1 may induce robust B-cell immunity in swine infected with ASFV.

## 4. Discussion

ASFV has rapidly spread across China since the first outbreak of the disease in the country in August 2018 [27]. The infection has high resulted in pig mortality rates as well as unprecedented and significant economic losses for the pig industry in the country. The development of a vaccine for ASFV is problematic due to the complex structure of ASFV and the complicated pathogenesis and immune response mechanisms associated with the virus. The p54 protein encoded by the E318L gene [28] possesses a transmembrane region near the N terminus, although the protein also appears in the replication factories of infected cells during ASFV replication [29]. The specific antibodies that detect p54 expression in Vero cells demonstrate that the protein activates caspase-3 during the early stages of ASFV infection and thus induces apoptosis, which is the first indication that ASFV induces cell death [30,31]. Compared with other structural proteins, p54 is present in the early stages of virus replication and plays a crucial role in viral replication, transfection, and maintaining structural stability. It is also the primary immunogenic protein of ASFV [26,28]. Amino acid sequence analysis reveals that p54 is a unique protein of ASFV and has no overlap with other viral protein sequences. The serodiagnosis of ASFV at present is mainly based on the antibody detection of the p30, p54, and p72 proteins [20,21,22,23,24,25,26,27,28,29,30,31,32]. Further screening of the main ASFV target proteins and the preparation of corresponding specific mAbs may help to provide more varied options for the establishment of serological assays against ASFV.

Serological diagnostic tests are valuable methods for identifying pathogens through the detection of antigens or antibodies. However, many pathogens, especially those that are similar, may have cross-reactions, making it difficult to clearly distinguish the course of the disease in serological assays. Epitope mapping technology has emerged to improve specificity and sensitivity and eliminate cross-reactions by designing epitope-based serological assays. Since epitopes specifically interact with chemical groups that play a role in antigenicity, their combination has the potential to distinguish different courses of the disease [33]. P54 is a structural protein of ASFV and is a type II transmembrane protein produced after infection. This study not only confirmed the immunogenicity of the P54 protein and the production of mAbs but also identified the linear epitopes located on the P54 protein.

Another approach involves subunit vaccines, which use specific proteins or peptides from the ASF virus to stimulate an immune response. The p72 protein, which is the major capsid protein of the ASF virus, has been the focus of many subunit vaccine studies. More recently, studies have also focused on the p30 and p54 proteins as potential vaccine targets. The p54 protein has been identified as the primary immunogenic protein of ASFV and has been studied as a potential vaccine candidate. Several studies have investigated the use of the recombinant p54 protein or p54-based subunit vaccines in experimental animal models. One study showed that the pigs vaccinated with a recombinant p54 protein vaccine were able to survive challenge with a virulent ASFV strain [34]. Another study found that pigs vaccinated with a p54-based subunit vaccine showed reduced clinical signs and viral load upon challenge with a virulent ASFV strain [35].

The identification of B-cell epitopes in viral structural proteins plays a crucial role in understanding virus–host interactions and developing effective diagnostic tools and vaccines, particularly for key antigenic proteins such as p30, p72, and p54. Monoclonal antibodies (mAbs) have high specificity and can be used to precisely locate antigen epitopes in viral proteins, making them a powerful tool in epitope mapping research. By identifying these epitopes, we can gain valuable insights into the immune response to ASFV and develop more targeted strategies for controlling and preventing ASFV infection. In this study, the antigenic epitopes of the four mAbs were localized, ten truncated peptides were used as antigens, and the four mAbs were used as primary antibodies. The P1 and P9 polypeptides identified by mAbs were described previously [36]. However, here, we discovered a novel linear B-cell epitope 110-TMSAIENLR-118 that was not identified previously. mAb 6E8G8E1 specifically recognized the truncated recombinant p54 protein and defined an epitope that was conserved in all the reference ASFV strains. This epitope is also conserved among ASFV wild strains and highly pathogenic epidemic strains from all regions of China. Thus, we speculate that this epitope may have significant value in ASFV vaccine design and development.

## Figures and Tables

**Figure 1 viruses-15-00867-f001:**
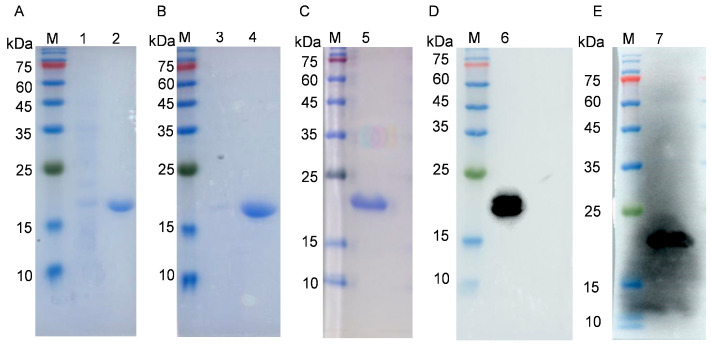
Identification of p54 by SDS-PAGE and Western blot detection of purified recombinant protein from *E. coli*: (**A**) lanes: M, protein molecular weight marker; 1, uninduced cells; 2, cells induced for p54 production; (**B**) lanes: M, protein molecular weight marker; 1, pellet fraction after lysis of cells induced for p54; 2, supernatant after lysis of cells induced for p54; (**C**,**D**) lanes: M, protein molecular weight marker; 1, purified p54; (**E**) lanes: M, protein molecular weight marker; 1, purified p54.

**Figure 2 viruses-15-00867-f002:**
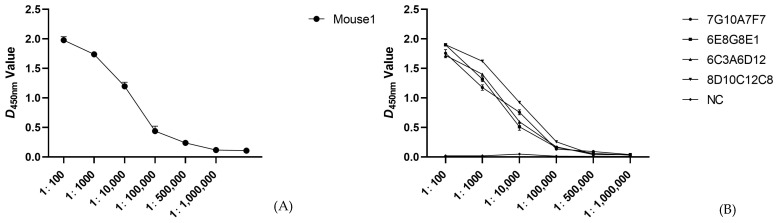
Detection of potency in mice (**A**) and monoclonal antibody potency (**B**). Data are presented as means ± standard deviations..

**Figure 3 viruses-15-00867-f003:**
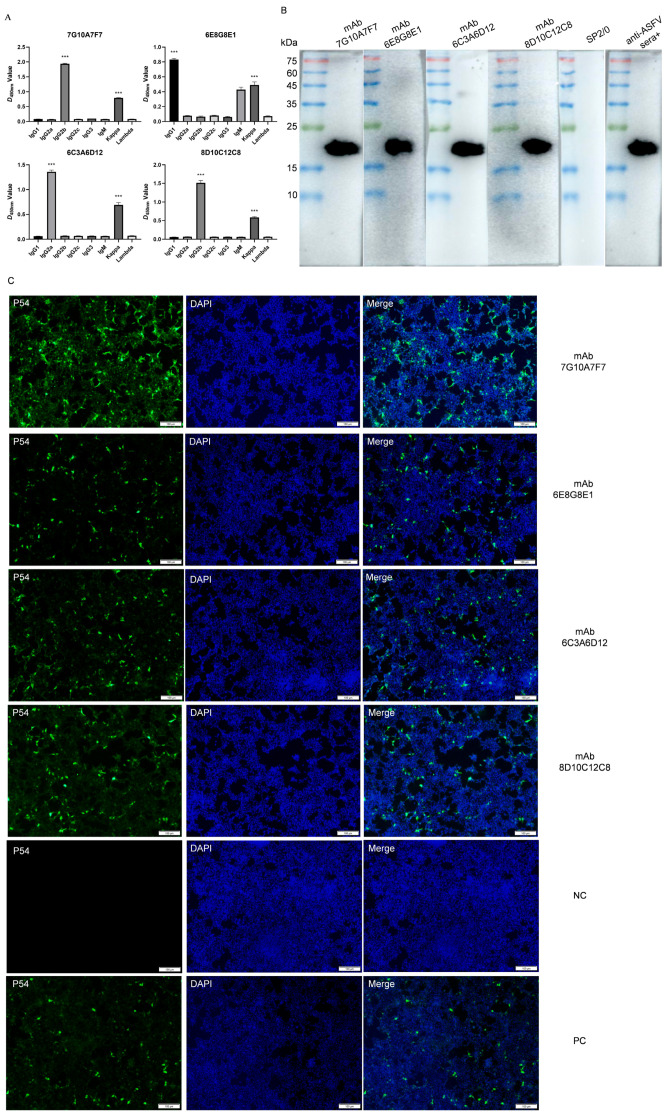
Characterization of mAbs against p54: (**A**) isotype determination of mAbs 7G10A7F7, 6E8G8E1, 6C3A6D12, and 8D10C12C8; (**B**) Western blotting results of mAbs against p54 protein. SP2/0 cell supernatants and positive anti-ASFV sera (anti-ASFV sera+) were used as negative and positive controls, respectively; (**C**) the reactivity of mAbs was analyzed with immunofluorescence assays. Cells were fixed at 48 hpi and incubated with hybridoma supernatants as the primary antibody and FITC-conjugated goat anti-mouse IgG as the secondary antibody. Data are presented as means ± standard deviations. The differences were set to be very significant at *** *p* < 0.001.

**Figure 4 viruses-15-00867-f004:**
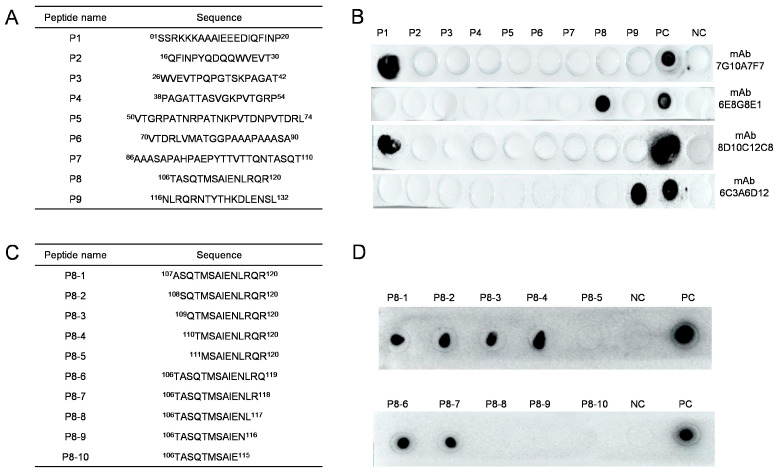
Identification of the epitopes recognized by mAbs 7G10A7F7, 6E8G8E1, 6C3A6D12, and 8D10C12C8: (**A**) nine synthetic overlapping peptides that span the P54 protein; (**B**) dot blot analysis of mAbs against the nine peptides; (**C**) ten shorter overlapping peptides that span peptide 8 (106-TASQTMSAIENLRQR-120); (**D**) dot blot analysis of mAb against the ten shorter peptides derived from peptide 8. PBS and purified p54 protein were used as negative and positive controls, respectively.

**Figure 5 viruses-15-00867-f005:**
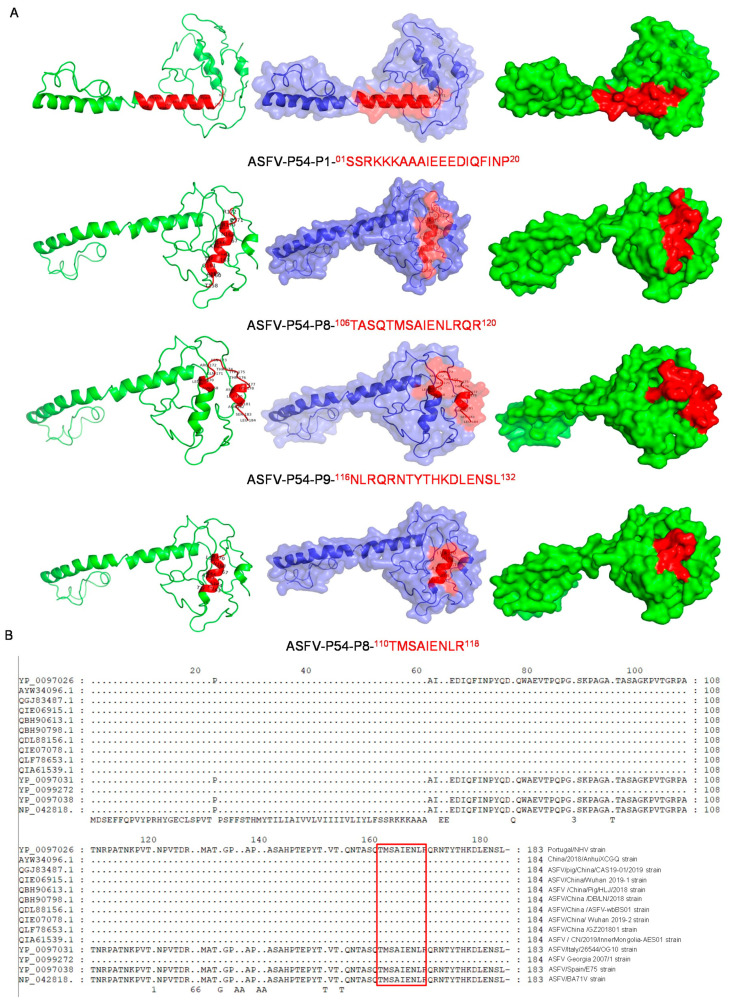
Bioinformatic analysis of p54 epitopes. Epitope 110-TMSAIENLR-118 was subjected to multiple sequence alignment with p54 from other ASFV strains and the spatial structure of the epitopes was screened: (**A**) epitopes P1 (01-SSRKKKAAAIEEEDIQFINP-20), P8 (106-TASQTMSAIENLRQR-120) (red), and P9 (116-NLRQRNTYTHKDLENSL-132) (red) are shown on the p54 protein; (**B**) the new p54 epitope 110-TMSAIENLR-118 (red) is displayed on P54.

**Figure 6 viruses-15-00867-f006:**
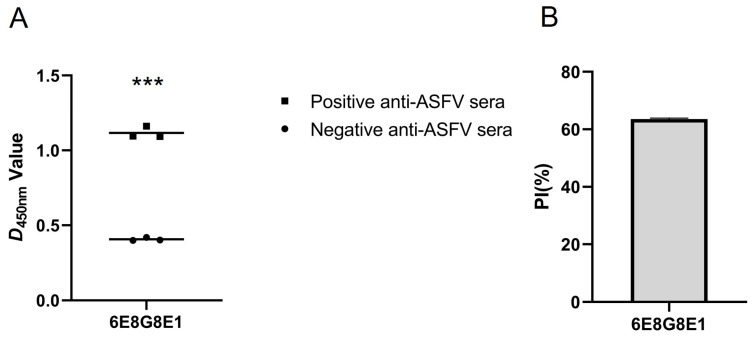
(**A**) Blocking/competitive ELISA detection of positive anti-ASFV sera against p54 mAbs; (**A**) negative anti-ASFV sera were used to determine baseline OD_450_ values in ELISA; (**B**) data are presented as means ± standard deviations. Statistical significance was determined by paired *t*-tests. *p* < 0.01; ***.

## Data Availability

All available data are presented in this manuscript.

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
