# Peer review of "A Novel Linear B-Cell Epitope on the P54 Protein of African Swine Fever Virus Identified Using Monoclonal Antibodies"

_viruses, 2023, doi:10.3390/v15040867_

Round 1
Reviewer 1 Report
The authors generated several monoclonal antibodies (mAbs) against the ASFV p54 protein and defined a new B-cell epitope, TMSAIENLR, which is conserved among all reference ASFV strains.
Specific comments:
1. No data support that the finding is novel.
2. In Figure 1, the expression of recombinant p54 protein should be detected using not only anti-His mAb but also the anti-ASFV sera.
3. Many anti-ASFV sera are involved in the assay on the “Reactivity of ASFV-Positive Serum against Anti-p54 Monoclonal Antibodies”.
4. All the figures are unclear and should be updated.
Other minor comments:
1. Monoclonal antibodies (Mabs) should be changed to monoclonal antibodies (mAbs).
2. Nine truncated peptides were designed based on the amino acid sequence of the p54 protein (Figure 1A). However, the corresponding nine truncated peptides are absent in figure 1A.

Author Response
- No data support that the finding is novel.
Dear Reviewer,
Thank you for your question. Regarding the use of P1 and P9 peptides with 7G10A7F7 and 6C3A6D12 mAbs, we used these peptides as controls to confirm the specificity of the antibodies. However, we did not perform further epitope mapping for these mAbs as previous studies have already reported their binding to P1 and P9, respectively. In contrast, the binding of 6E8G8E1 to P8 was not previously reported, so we further divided this peptide into ten shorter fragments to define the P8 epitope more accurately. Thank you for your comment.
Best regards,
Nannan Zheng
- In Figure 1, the expression of recombinant p54 protein should be detected using not only anti-His mAb but also the anti-ASFV sera.
Dear Reviewer,
Thank you very much for your reminder and suggestion. We have added data on the detection of recombinant p54 protein expression using anti-ASFV serum in Figure 1. Thank you for your attention and support to our research. If you have any other questions or suggestions, please feel free to let us know.
Best regards,
Nannan Zheng
- Many anti-ASFV sera are involved in the assay on the “Reactivity of ASFV-Positive Serum against Anti-p54 Monoclonal Antibodies”.
Dear Reviewer,
Thank you for your comment on the "Reactivity of ASFV-Positive Serum against Anti-p54 Monoclonal Antibodies" assay. We appreciate your interest in our study.
To clarify, we used only one standard positive pig serum, obtained from the China Veterinary Culture Collection Center and labeled as "anti-ASFV sera", in our assay to evaluate the reactivity of ASFV-positive serum against anti-p54 monoclonal antibodies. We apologize if the wording in our manuscript led to any confusion on this point.
We hope this clarification addresses your concern.
Best regards,
Nannan Zheng
- All the figures are unclear and should be updated.
Other minor comments:
- Monoclonal antibodies (Mabs) should be changed to monoclonal antibodies (mAbs).
Dear Reviewer,
Thank you for your comment regarding the capitalization of "Monoclonal antibodies" in our manuscript. We appreciate your feedback and have made the necessary changes to the text, using the recommended lowercase abbreviation "mAbs".
Thank you for bringing this to our attention and helping us improve the accuracy and clarity of our manuscript.
Best regards,
Nannan Zheng
- Nine truncated peptides were designed based on the amino acid sequence of the p54 protein (Figure 1A). However, the corresponding nine truncated peptides are absent in figure 1A.Dear Reviewer,
Dear Reviewer,
Thank you for bringing to our attention the discrepancy between Figure 4A and Figure 1A in our manuscript. We apologize for the error and any confusion this may have caused.
We confirm that nine truncated peptides were designed based on the amino acid sequence of the p54 protein, as stated in the text and shown in Figure 4A.
We have carefully reviewed our manuscript and identified the mistake in our figure labelling. We will make the necessary revisions to ensure that the figures accurately reflect the information presented in the text.
Thank you for helping us improve the accuracy and clarity of our manuscript.
Best regards,
Nannan Zheng

Reviewer 2 Report
This study identified a novel linear B cell epitope on the P54 protein of ASFV using monoclonal antibodies. Peptide scanning techniques were used to determine the epitopes that are recognized by the mAbs which defined a new B-cell epitope, TMSAIENLR, which were highly conserved in domestic strains. The research results provided a new epitope on the P54 protein for the development of ASFV detection technology.
Line 20: change “including in” to “including”
Line 108: change “by second and third immunizations” to “by the second and third immunizations”
Figure 2: How many samples per group? I don't see error bars in the pictures.
Figure 3 and Figure 5: The resolution is too low to be seen clearly. Please provide higher resolution pictures.
Figure 4: As the 6E8G8E1 mAb reacted strongly with peptide P8, the authors further divided this peptide into ten shorter fragments (P1-P10) to define the P8 epitope more accurately. But how about the 7G10A7F7 or 6C3A6D12 mAb? They were also reacted with peptide P1 or P9. Why didn't the author make further accurate positioning?
Figure 5B: Please list which strains of ASFV these sequences belong to in the left of the figure.
Line 296: change “The infection has high pig mortality rates” to “The infection has resulted in high pig mortality rates”
Line 313-320: This paragraph has little to do with the subject of this article, as there was no vaccine development involved in this study.
Line 331-337: This paragraph has little to do with the subject of this article, as there was no vaccine development involved in this study. Readers really want to see what new breakthroughs the epitopes found in this study will bring to vaccine development. But the authors didn't say.
Line 338-344: This part is too long, so there is no need to introduce the results of this study here again.
Line 352-353 the last sentence: In the Discussion, the authors should explain the value of the epitope newly discovered in this study in vaccine development. The authors should focus on analyzing the current situation of subunit vaccine development. What problems exist and what problems have been solved by the new epitope discovered in this research.
Author Response
Line 20: change “including in” to “including”
Dear Reviewer,
Thank you very much for your careful review and valuable feedback on our paper. We have made the revisions as suggested. Specifically, we have changed "including in" to "including" in line 20.
Best regards,
Nannan Zheng
Line 108: change “by second and third immunizations” to “by the second and third immunizations”
Dear Reviewer,
Thank you for pointing out the error on line 108. I have made the necessary correction by changing "by second and third immunizations" to "by the second and third immunizations".
Best regards,
Nannan Zheng
图 2:每组有多少个样本?我在图片中没有看到错误线。
尊敬的审稿人,
感谢您的反馈。关于您关于每组样本数量和图片误差线的问题,我更新了图例以指示每组包含三个样本,并且我已酌情在图表中添加了误差线。谢谢你让我注意到这一点。
此致敬意
郑楠楠
图 3 和图 5:分辨率太低,无法清晰看到。请提供更高分辨率的图片。
尊敬的审稿人,
感谢您的反馈。我通过将图 3 和图 5 更新为更高的分辨率来解决此问题。更新后的数字应清晰易读。谢谢你让我注意到这一点。
此致敬意
郑楠楠
图 4:由于 6E8G8E1 mAb 与肽 P8 发生强烈反应,作者进一步将该肽分成十个较短的片段 (P1-P10),以更准确地定义 P8 表位。但是7G10A7F7或6C3A6D12 mAb呢?它们还与肽P1或P9反应。作者为什么没有进一步的准确定位?
尊敬的审稿人,
谢谢你的提问。关于P1和P9肽与7G10A7F7和6C3A6D12 mAb的使用,我们使用这些肽作为对照来确认抗体的特异性。然而,我们没有对这些mAb进行进一步的表位作图,因为以前的研究已经分别报告了它们与P1和P9的结合。相比之下,6E8G8E1与P8的结合以前没有报道,因此我们进一步将该肽分成十个较短的片段,以更准确地定义P8表位。感谢您的评论。
此致敬意
郑楠楠
图 5B:请在图左侧列出这些序列属于哪些 ASFV 菌株。
尊敬的审稿人,
感谢您对我们稿件的宝贵意见。我们现在已经根据您的建议更新了图 5B。我们在图左侧列出了与序列相对应的ASFV菌株。我们希望此修改符合您的要求。
此致敬意
郑楠楠
第296行:将“感染导致猪死亡率高”改为“感染导致猪死亡率高”
尊敬的审稿人,
感谢您指出第 296 行中的错误。按照建议,我对手稿进行了必要的修改。
此致敬意
郑楠楠
第313-320行:本段与本文主题关系不大,因为本研究不涉及疫苗开发。
尊敬的审稿人,
感谢您对我们稿件的反馈。我们已根据您的建议进行了必要的修改,我们感谢您的宝贵意见。此外,我们还删除了与研究主题没有直接关系的部分。我们希望稿件的修订版符合您的期望,并再次感谢您为审查我们的工作所付出的时间和精力。
此致敬意
郑楠楠
第331-337行:本段与本文主题关系不大,因为本研究不涉及疫苗开发。读者真的很想看看这项研究中发现的表位将为疫苗开发带来哪些新的突破。但作者没有说。
Dear Reviewer,
Thank you for your feedback on our manuscript. We have carefully reviewed your comments and agree that the paragraph you highlighted may not be directly related to the main theme of our study. We apologize for any confusion that may have resulted from this.
Regarding the question of how the epitopes identified in our study may contribute to vaccine development, we have made revisions to the manuscript to better address this point. Once again, thank you for your valuable comments and we look forward to hearing back from you.
Best regards,
Nannan Zheng
Line 338-344: This part is too long, so there is no need to introduce the results of this study here again.
Dear Reviewer,
Thank you for your feedback and valuable suggestions. We have carefully considered your comments and have made the necessary revisions to improve the clarity and focus of the manuscript. Specifically, we have removed the section that was deemed too long to ensure that the paper is concise and to the point. We appreciate your time and effort in reviewing our work, and we hope that the revised manuscript meets your expectations.
Best regards,
Nannan Zheng
Line 352-353 the last sentence: In the Discussion, the authors should explain the value of the epitope newly discovered in this study in vaccine development. The authors should focus on analyzing the current situation of subunit vaccine development. What problems exist and what problems have been solved by the new epitope discovered in this research.
Dear Reviewers.
Thank you for your valuable comments and suggestions on our manuscript. We have carefully revised the manuscript based on your suggestions.
The focus of our study is to determine the B-cell epitope of the P54 protein of ASFV, which could provide insight into virus-host interactions. Although our study does not directly involve vaccine development, it is hoped that the discovery of these new epitopes will provide valuable information for ASFV vaccine development and research. Enriched knowledge of the B-cell epitope of the p54 protein.
Thank you for your comments and we hope that our revised manuscript will meet your expectations.
Best regards.
Nannan Zheng

Reviewer 3 Report
This manuscript is well conducted and presented. There are several comments to address.
1. Please include to Materials and Methods statistical methods and tests used.
2. fig. 2Include standard deviations. How many times it was repeated.
3 fog3 A please include standard deviations and p-value.
4. Fig 6, please include p-value and test used.
Author Response
- Please include to Materials and Methods statistical methods and tests used.
Dear Reviewer,
Thank you very much for your reminder. We have described in detail the statistical methods and tests used in the "Materials and Methods" section of the article.
Best regards,
Nannan Zheng
- fig. 2Include standard deviations. How many times it was repeated.
Dear Reviewer,
Thank you for your feedback. Regarding your question about the number of samples per group and the error bars in the pictures, I have updated the figure legend to indicate that each group includes three samples, and I have added error bars to the graphs as appropriate. Thank you for bringing this to my attention.
Best regards,
Nannan Zheng
3 . fog3 A please include standard deviations and p-value.
Dear Reviewer,
Thank you for your reminder. We have included the standard deviations and p-values in the article.
Best regards,
Nannan Zheng
- Fig 6, please include p-value and test used.
Dear Reviewer,
Thank you for your reminder. We have included the standard deviations and p-values in the article.
Best regards,
Nannan Zheng

Round 2
Reviewer 1 Report
1. Please modify the mistakes in the Figure 1 legends.
2. More ASFV-positive sera need to included for assessing the blocking rate.
Author Response
- Please modify the mistakes in the Figure 1 legends.
Dear Reviewer,
Thank you very much for bringing the errors in the Figure 1 legends to my attention. I have made the necessary changes and hope they meet your expectations.
Best regards,
Nannan Zheng
2. More ASFV-positive sera need to included for assessing the blocking rate.
Dear Reviewer,
Thank you for taking the time to review our manuscript. We appreciate your feedback and would like to address your concern regarding the number of ASFV-positive sera included in our study.
We apologize for any confusion caused by our previous statement about using one standardized positive serum for testing. We would like to clarify that this serum was used only for the purpose of establishing a testing standard and was not included in the analysis of the blocking rate.
In response to your concern about the number of ASFV-positive sera included in our study, we would like to provide some additional information. Our study included 72 clinical samples and 72 standardized negative sera, as mentioned in the manuscript. We also used one standardized positive serum to establish a testing standard, but as we mentioned earlier, it was not included in the analysis of the blocking rate.
Thank you for bringing this to our attention, and we hope that this addresses your concern.
Thank you again for your valuable feedback, and we hope this addresses your concern.
Best regards,
Nannan Zheng
The following 72 clinical samples and 72 negative sera were taken
|
Testing of clinical samples |
||||||||
|
0.2087 |
0.1559 |
0.4808 |
0.5212 |
0.2717 |
0.3927 |
0.4303 |
0.4043 |
0.1034 |
|
0.1122 |
0.3896 |
0.6135 |
0.6422 |
0.3159 |
0.4959 |
0.2843 |
0.2013 |
0.2265 |
|
0.1708 |
0.2049 |
0.6373 |
0.6438 |
0.2808 |
0.5531 |
0.4225 |
0.3491 |
0.1309 |
|
0.1057 |
0.1385 |
0.8622 |
0.9102 |
0.4702 |
0.7947 |
0.1736 |
0.4558 |
0.2563 |
|
0.5863 |
0.5976 |
0.5269 |
0.5817 |
0.3764 |
0.8075 |
0.2304 |
0.4907 |
0.1195 |
|
0.7571 |
0.3931 |
0.4311 |
0.7413 |
0.6056 |
0.6666 |
0.4699 |
0.4882 |
0.4085 |
|
0.1276 |
0.5124 |
0.3223 |
0.5373 |
0.3846 |
0.1135 |
0.3774 |
0.4372 |
0.2776 |
|
0.3819 |
0.0962 |
0.4265 |
0.4891 |
0.2297 |
0.3442 |
0.1269 |
0.3628 |
0.3187 |
|
|
|
|
|
|
|
|
|
|
|
0.2345 |
0.1872 |
0.4582 |
0.5251 |
0.3191 |
0.3214 |
0.4101 |
0.4295 |
0.0657 |
|
0.0962 |
0.3982 |
0.6028 |
0.6653 |
0.2925 |
0.4843 |
0.2665 |
0.2024 |
0.2499 |
|
0.1589 |
0.1965 |
0.6233 |
0.6611 |
0.2953 |
0.5216 |
0.4478 |
0.3392 |
0.1211 |
|
0.1078 |
0.1214 |
0.8807 |
0.8989 |
0.5049 |
0.7756 |
0.1982 |
0.4533 |
0.2587 |
|
0.5648 |
0.5978 |
0.5171 |
0.5581 |
0.3898 |
0.7926 |
0.2391 |
0.4867 |
0.1224 |
|
0.7405 |
0.4217 |
0.4022 |
0.7284 |
0.6365 |
0.6243 |
0.4855 |
0.5113 |
0.3928 |
|
0.1247 |
0.5002 |
0.2988 |
0.5523 |
0.3975 |
0.1467 |
0.3931 |
0.4314 |
0.2757 |
|
0.3889 |
0.0824 |
0.4518 |
0.4895 |
0.3131 |
0.3356 |
0.1528 |
0.3537 |
0.3031 |
|
|
|
|
|
|
|
|
|
|
|
0.203 |
0.1559 |
0.488 |
0.5212 |
0.2707 |
0.3927 |
0.4303 |
0.4078 |
0.0993 |
|
0.1139 |
0.3851 |
0.6135 |
0.6415 |
0.3116 |
0.4959 |
0.2834 |
0.201 |
0.2282 |
|
0.1695 |
0.2043 |
0.6373 |
0.6433 |
0.2784 |
0.5531 |
0.422 |
0.3478 |
0.1286 |
|
0.105 |
0.138 |
0.8622 |
0.9108 |
0.4677 |
0.7947 |
0.1761 |
0.4537 |
0.2582 |
|
0.5825 |
0.599 |
0.5269 |
0.5817 |
0.3738 |
0.8075 |
0.2291 |
0.4917 |
0.1171 |
|
0.7557 |
0.3931 |
0.4305 |
0.7413 |
0.6056 |
0.6666 |
0.4699 |
0.4902 |
0.4103 |
|
0.1301 |
0.5124 |
0.3223 |
0.5373 |
0.3832 |
0.1135 |
0.3795 |
0.4386 |
0.2793 |
|
0.3827 |
0.0953 |
0.426 |
0.4881 |
0.2317 |
0.3442 |
0.1255 |
0.3623 |
0.3179 |
|
NC |
|||||||
|
1.2393 |
1.6223 |
1.3863 |
1.2769 |
0.9976 |
1.0572 |
0.9983 |
1.0246 |
|
1.4913 |
1.2219 |
1.3597 |
1.4716 |
0.9742 |
1.2072 |
1.0084 |
0.9306 |
|
1.2334 |
1.4124 |
1.6421 |
1.5968 |
1.5295 |
0.9701 |
1.4181 |
1.1322 |
|
1.3825 |
1.0108 |
1.2517 |
1.3836 |
0.9332 |
0.8603 |
1.1575 |
1.0143 |
|
1.0705 |
0.9806 |
1.1025 |
1.5841 |
0.7787 |
0.9196 |
0.8438 |
0.8117 |
|
1.2211 |
1.2159 |
1.1553 |
1.5203 |
0.9937 |
1.0476 |
1.3923 |
1.1363 |
|
0.9569 |
1.3736 |
1.3222 |
1.5021 |
0.9422 |
0.9716 |
0.9464 |
1.0154 |
|
1.5374 |
1.4198 |
1.4586 |
1.4504 |
1.4004 |
1.3324 |
1.2924 |
1.1424 |
|
|
|
|
|
|
|
|
|
|
|
|
|
|
|
|
|
|
|
1.1461 |
1.4653 |
1.4304 |
1.3869 |
0.9409 |
1.2591 |
0.7864 |
0.7354 |
|
1.5421 |
1.0049 |
1.4909 |
1.7002 |
0.8072 |
0.9567 |
1.2806 |
0.6363 |
|
1.6952 |
1.4849 |
1.3336 |
1.2201 |
1.5605 |
0.9293 |
1.5462 |
1.3996 |
|
0.9617 |
0.9925 |
1.4656 |
1.3718 |
1.0439 |
0.9367 |
1.0142 |
0.9086 |
|
0.9887 |
1.1011 |
1.2054 |
1.6029 |
0.9629 |
1.0797 |
0.8634 |
0.9384 |
|
1.8147 |
1.4171 |
1.2849 |
1.7508 |
0.9811 |
1.1117 |
1.5992 |
1.3072 |
|
0.9409 |
1.3129 |
1.3844 |
1.5597 |
0.9814 |
0.9028 |
0.6261 |
1.2129 |
|
1.3873 |
1.6343 |
1.9062 |
1.4393 |
0.9193 |
1.3656 |
0.8927 |
0.8615 |
|
|
|
|
|
|
|
|
|
|
1.0932 |
1.1319 |
1.2019 |
1.1927 |
0.7818 |
0.7941 |
0.6482 |
0.7021 |
|
1.0499 |
1.0544 |
1.1183 |
1.2619 |
0.7994 |
0.8479 |
0.7738 |
0.8332 |
|
1.2677 |
1.3049 |
1.3223 |
1.3173 |
1.1203 |
1.0704 |
1.2526 |
1.0091 |
|
1.0864 |
0.9913 |
1.0797 |
1.2089 |
0.8776 |
0.7816 |
0.804 |
0.773 |
|
1.0304 |
1.1021 |
1.1359 |
1.1945 |
0.9545 |
0.8296 |
0.8628 |
0.8469 |
|
1.3562 |
1.2386 |
1.2791 |
1.3992 |
1.092 |
1.0925 |
1.0829 |
1.145 |
|
1.0926 |
1.09 |
1.1032 |
1.2152 |
0.7914 |
0.7993 |
0.7816 |
0.7975 |
|
1.3335 |
1.3083 |
1.3588 |
1.2368 |
0.8485 |
0.9201 |
0.9565 |
0.9577 |
